# Correlation Relationship between Phase Inversion of Pickering Emulsions and Biocatalytic Activity of Microbial Transformation of Phytosterols

Wenyu Zhao [1], Haisheng Xie [1], Xuehong Zhang [2] and Zhilong Wang [1,*]

[1]  State Key Laboratory of Microbial Metabolism and Engineering Research Center of Cell & Therapeutic Antibody, Ministry of Education, School of Pharmacy, Shanghai Jiao Tong University, 800 Dongchuan, Shanghai 200240, China
[2]  State Key Laboratory of Microbial Metabolism, School of Life Science and Biotechnology, Shanghai Jiao Tong University, 800 Dongchuan, Shanghai 200240, China
*  Correspondence: zlwang@sjtu.edu.cn

**Abstract:** Microbial transformation of hydrophobic phytosterols into the pharmaceutical steroid precursors AD (androst-4-ene-3, 17-dione) and ADD (androst-4-diene-3, 17-dione) in a water–plant oil two-phase system by *Mycolicibacterium neoaurum* is a paradigm of interfacial biocatalysis in Pickering emulsions stabilized by bacterial cells. In the present work, phase inversion of Pickering emulsions—i.e., Pickering emulsions turning from water-in-oil (W/O) emulsions into oil-in-water (O/W) ones—was observed during microbial transformation in the presence of high concentrations of crystal phytosterols. It was found that there is a correlation relationship between the phase behaviors of Pickering emulsions and the biocatalytic activity of utilizing *M. neoaurum* as a whole-cell catalyst. Efficient microbial transformation under the high crystal phytosterol loadings was achieved due to the formation of O/W emulsions where interfacial biocatalysis took place. Under the optimal conditions (volume ratio of soybean oil to water: 15:35 mL, phytosterols concentration in the soybean oil: 80 g/L, glucose as co-substrate in the aqueous culture medium: 10 g/L), the concentrations of AD and ADD reached 4.8 g/L based on the whole broth (16 g/L based on the oil phase) after microbial transformation for 9 days.

**Keywords:** microbial transformation; phytosterols; Pickering emulsion; phase inversion; plant oil

## 1. Introduction

Using *Mycolicibacterium neoaurum* as whole-cell biocatalyst, microbial transformation of phytosterols into the corresponding steroid precursors—such as AD (androst-4-ene-3, 17-dione) and ADD (androst-4-diene-3, 17-dione)—is an important route in the steroid pharmaceutical industry [1–4]. The industrial process is a paradigm for microbial transformation of crystal substrates due to the limited solubility of phytosterols in aqueous solutions, such as the solubility of β-sitosterol—a major component of phytosterols (approximately 2 mg/L in an aqueous solution [5], approximately 68 mM (28 g/L) in BEHP (bis-(2-ethylhexyl) phthalate) [6])—and the solubility of soybean sterols (approximately 13 g/L in soybean oil and 8 g/L in coconut oil [5]). There are two major strategies for enhancing mass transport during microbial transformation of phytosterols. One method is the solubilization of phytosterols in an aqueous solution by addition of biocompatible solubilizers, such as hydroxypropyl-β-cyclodextrin [7,8]. Another is microbial transformation in a water–organic solvent two-phase system [4,5]. In this case, it is believed that the organic solvent phase acts as a reservoir of phytosterols and microbial transformation takes place in the aqueous solution phase, where diffusion of phytosterols from the organic solvent phase to the aqueous solution is more effective than that from phytosterol crystals to the aqueous solution [9]. Many other two-phase systems have also been exploited for microbial transformation of phytosterols, such as water–ionic liquid two-phase systems [10,11],

polymer-based aqueous two-phase systems [12], and nonionic-surfactant-based cloud point systems [13,14].

Rather than focusing on the diffusion of phytosterols in the water–organic solvent two-phase system, the location of bacterial cells in the medium may be more important. Recently, a novel mechanism for *M. neoaurum* to utilize phytosterols solubilized in the BEHP (bis-(2-ethylhexyl) phthalate) phase of a water–BEHP two-phase system has been proposed. During microbial culture using cholesterol as the sole carbon source, the water–BEHP two-phase system turns into Pickering emulsions, where the hydrophobic bacterial cells act not only as solid emulsifiers by attachment on the oil–water interfaces, but also as biocatalysts for microbial transformation of cholesterol into AD and ADD [15]. This process is known as interfacial biocatalysis, because the bacteria attached to the oil–water interfaces assimilate both hydrophobic nutrients solubilized in the oil phase and hydrophilic ones solubilized in the water phase. By loading pre-cultured bacterial cells in the water–BEHP two-phase system (i.e., bacterial-cell-stabilized Pickering emulsions), microbial transformation of phytosterols also produces AD and ADD successfully [4]. However, both microbial growth and microbial transformation to produce AD and ADD in the water–BEHP two-phase system fail to utilize phytosterols as the sole carbon source. Conversely, the addition of glucose as a co-substrate for microbial growth and microbial transformation of phytosterols successfully produces AD and ADD in the bacterial-cell-stabilized Pickering emulsions [16]. This demonstrates that a certain amount of bacterial cells acting as solid emulsifiers is necessary for the formation of bacterial-cell-stabilized Pickering emulsions and, subsequently, the efficient microbial transformation of phytosterols to produce AD and ADD via interfacial biocatalysis. At the same time, crystal substrate inhibition—i.e., decreasing phytosterol degradation with the increase in crystal phytosterol loadings—is also observed during microbial transformation of phytosterols in the bacterial-cell-stabilized Pickering emulsions [16]. Unfortunately, the crystal substrate inhibition severely limits the efficient microbial transformation under the condition of high phytosterol loading.

Plant oils have been applied widely as natural solvents for extractive microbial transformation or fermentation due to their biocompatibility and low cost [17,18]. Water–plant oil two-phase systems have also been studied extensively for the microbial transformation of phytosterols in both academic research [5,19–22] and industrial application. In the present work, the influence of operational parameters (such as phytosterol concentrations, volume ratio of oil to water, and glucose concentration) on the basic structures of Pickering emulsions (such as the fraction of emulsion and the microscopic morphology) was investigated by using the microbial transformation of phytosterols in a water–soybean oil two-phase system as an example. Keeping interfacial biocatalysis in Pickering emulsions [15,16] in mind, the relationship between the emulsion structures and the bioactivity of microbial transformation under various conditions—especially with different phytosterol loadings—was correlated.

## 2. Results

### 2.1. Plant Oil Acting Both as Co-Substrate and as Solvent

Microbial culture was carried out in the aqueous culture medium in the absence of glucose while in the presence of various volumes of soybean oil as the sole carbon source (Figure 1A). It was observed that the biomass increased with the increase in soybean oil volume, indicating that the soybean oil served as the sole carbon source for microbial growth. In other words, microbial growth occurred in the water–soybean oil two-phase system. Furthermore, microbial transformation was also carried out in the water–soybean oil two-phase system by the addition of phytosterols (no phytosterol crystal was observed with 25 g/L of phytosterols in soybean oil) (Figure 1B). Phytosterols were converted into the corresponding AD and ADD. Meanwhile, the water–soybean oil two-phase system turned into W/O Pickering emulsions (data not shown). The microbial growth and microbial transformation occurred at the same time in the absence of glucose as a co-substrate. The

phenomenon was the same as the microbial transformation of phytosterols in a water–BEHP two-phase system by the addition of glucose as a co-substrate for microbial growth [16].

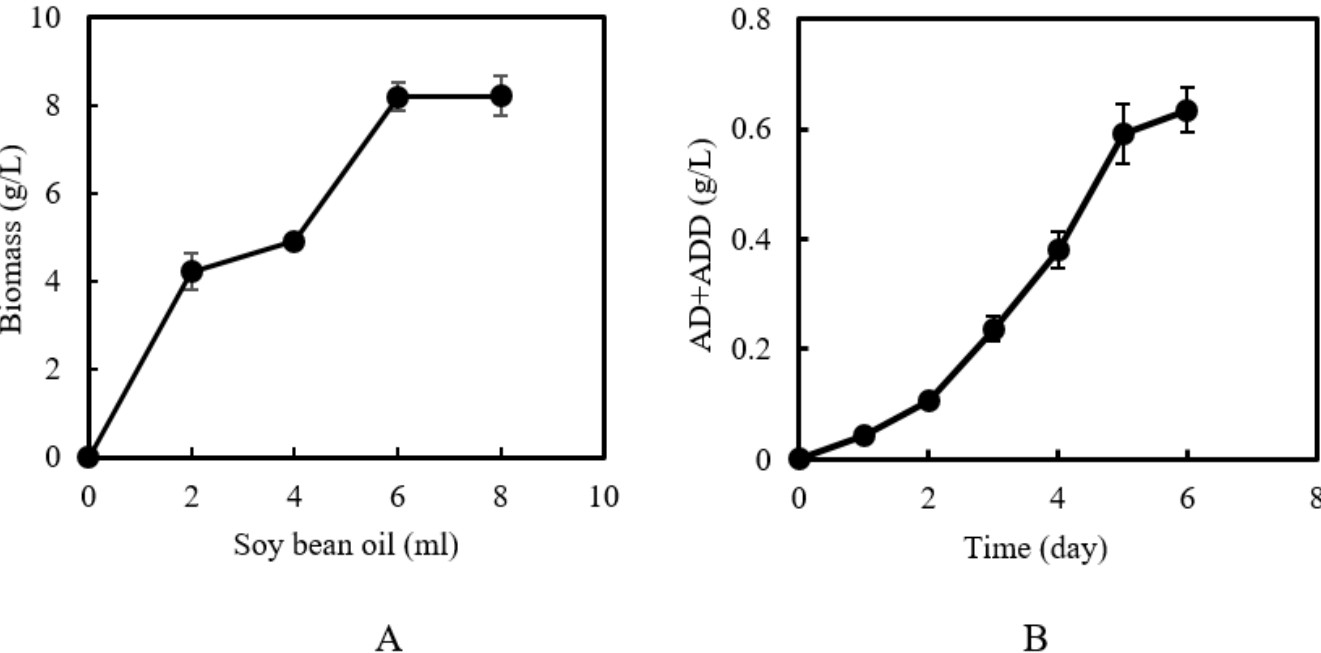

**Figure 1.** Soybean oil acting as both a substrate for microbial growth and a solvent for solubilized phytosterols: (**A**) Microbial culture with soybean oil as the sole carbon source. The microbial culture was carried out in the aqueous culture medium (40 mL) in the absence of glucose and with various volumes of soybean oil for 6 days. (**B**) Time course of microbial transformation of phytosterols in a water–soybean oil two-phase system. The basic condition was the aqueous culture medium in the absence of glucose (40 mL) and soybean oil (10 mL) with 25 g/L phytosterols.

### 2.2. Effect of Volume Ratio of Oil to Water

The plant oil, acting as both a co-substrate for microbial growth and an organic solvent for Pickering emulsions, influenced the structures of the Pickering emulsions and further influenced the accumulation of AD and ADD during microbial transformation in a water–soybean oil two-phase system (Figure 2). After microbial transformation, the water–soybean oil two-phase system turned into W/O Pickering emulsions. With the increase in the soybean oil volume from 5 to 15 mL (No. 1 to 3), the volume of the emulsion phase increased (i.e., the volume of the excess water phase decreased). The Pickering emulsions became very viscous with the increase in the soybean oil volume. With further increases in the oil volume (No. 4 and 5), the whole water–soybean oil two-phase system was emulsified completely, and no excess water phase was observed (Figure 2A). However, the flowability of the Pickering emulsions improved markedly at a high volume ratio of oil to water (data no shown, only direct observation of the shaken flasks during microbial transformation). Correspondingly, the microscopic morphology of the Pickering emulsions indicated that the sizes of the droplets decreased with the increase in the volume ratio of oil to water, i.e., relatively large droplets at a low volume ratio of oil to water (No. 1–3) but small droplets at a high volume ratio of oil to water (No. 5) (Figure 2B). The changes in the emulsion structures dramatically influenced the microbial transformation of the phytosterols (Figure 2C). The concentrations of AD and ADD increased dramatically with the increase in the oil volume (the concentration of phytosterols in the oil was kept at 25 g/L) and then reached their maximum of 2.1 g/L at the oil volume of 15 mL (No. 3), while further increases in the oil volume led to decreases in the AD and ADD concentrations.

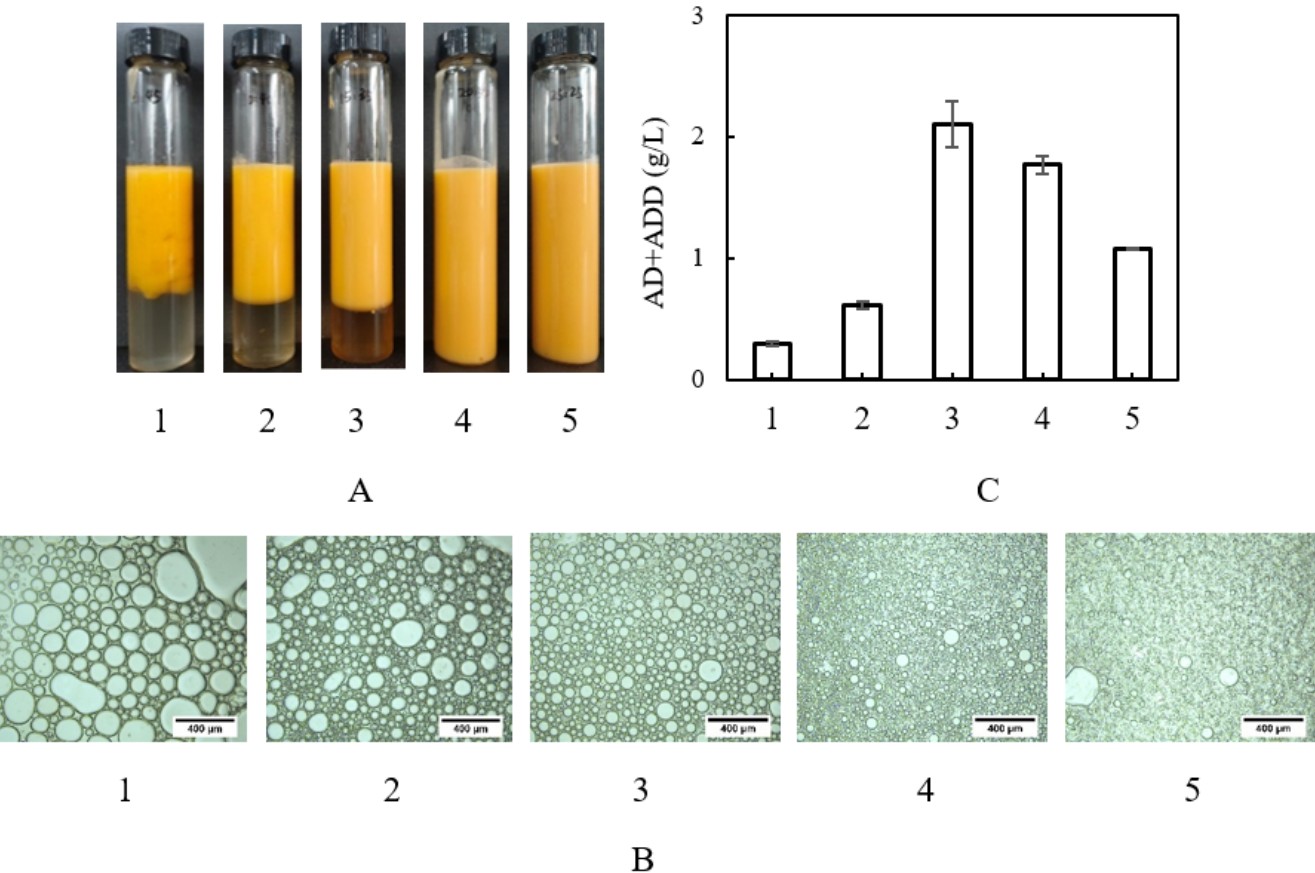

**Figure 2.** Effects of the volume ratio of soybean oil to water: (**A**) Emulsion fraction. (**B**) Microscopic emulsions (bar = 400 μm). (**C**) AD and ADD concentrations. In the basic experimental condition, the total volume of the soybean oil and the aqueous culture medium was 50 mL. There was no glucose in the aqueous culture medium. The concentration of phytosterols in the soybean oil was 25 g/L. The microbial transformation time was 7 days. No. 1, 2, 3, 4, and 5 corresponded to volume ratios of soybean oil to the aqueous culture medium of 5:45, 10:40, 15:35, 20:30, and 25:25 mL, respectively.

*2.3. Influence of Glucose as a Co-Substrate*

Glucose acting as a co-substrate enhanced the microbial growth, which influenced the Pickering emulsion structures and the subsequent accumulation of AD and ADD (Figure 3). Excess water phase was observed in the Pickering emulsions in the absence of glucose (No. 1). With the increase in the glucose concentration, the excess water phase disappeared (No. 2–4). Further increases in the glucose concentration caused the excess water phase to appear again (No. 5), whereupon the flowability of the emulsions became very good (Figure 3A). Microscopic observation showed that homogeneous small water droplets were present at the glucose concentration of 10 g/L (No. 2), while further increases in the glucose concentration led to the formation of heterogeneous irregular water droplets (No. 3 and 4). In particular, multiple oil-in-water-in-oil (O/W/O) emulsions were formed at the glucose concentration of 60 g/L (No. 5), where the bacterial cells were dispersed in the oil phase (Figure 3B). Correspondingly, the accumulation of AD and ADD achieved the maximum of 2.5 g/L at a glucose concentration of 10 g/L. Both the absence of glucose and high glucose concentrations were unfavorable for the accumulation of AD and ADD (Figure 3C).

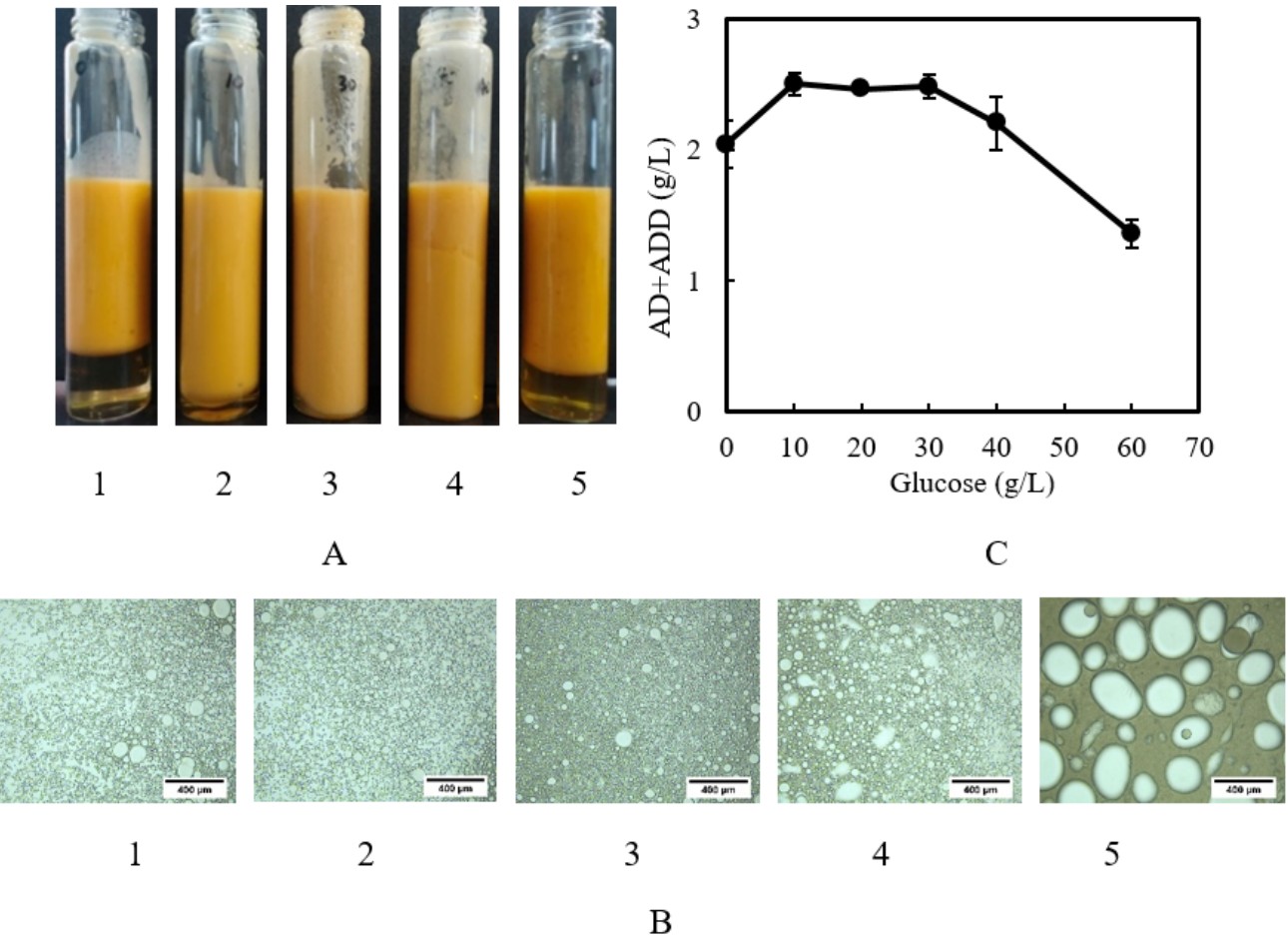

**Figure 3.** Influence of glucose acting as a co-substrate: (**A**) Emulsion fraction. (**B**) Microscopic emulsions (bar = 400 μm). (**C**) AD and ADD concentration. In the basic experimental condition, the volume ratio of soybean oil to the aqueous culture medium was 15:35 mL. The concentration of phytosterols in the soybean oil was 25 g/L. The microbial transformation time was 7 days. No. 1, 2, 3, 4, and 5 corresponded to glucose concentrations in the aqueous culture medium of 0, 10, 30, 40, and 60 g/L, respectively.

### 2.4. Microbial Transformation of Crystal Phytosterols

Under the optimal volume ratio of oil to water (15:35 mL) and glucose concentration (10 g/L) in the aqueous culture medium, the effect of phytosterol loading on the Pickering emulsion structures, and then on the accumulation of AD and ADD, was examined (Figure 4). No phytosterol crystals were observed at phytosterol concentrations below 25 g/L in the soybean oil. No excess water phase was observed when the phytosterol concentration was 10, 25, or 40 g/L (No. 1–3), while the flowability of the Pickering emulsions improved markedly at the phytosterol concentration of 40 g/L (No. 3). Further increases in the phytosterol concentration caused an excess water phase to appear, where some microaggregates of phytosterol crystals were observed with the adsorbed bacterial cells. The formation of bacteria–phytosterol crystal microaggregates was confirmed by confocal laser scanning microscopy (CLSM) in our previous work [16]. At the same time, the flowability of the Pickering emulsions further increased (No. 4 and 5) (Figure 4A). The microscopic morphology indicated that some homogeneous small water droplets were observed at the low phytosterol concentrations where no phytosterol crystals were observed (No. 1 and 2). Some heterogeneous irregular water droplets were observed and some bacterial cells were partitioned in the continuous oil phase at the phytosterol concentration of 40 g/L (No. 3). When increasing the phytosterol concentration to 50 g/L (No. 4), multiple accompanying O/W/O emulsions with good flowability low stability and were found, characteristic of

the phase inversion of Pickering emulsions. Subsequently, the Pickering emulsions turned from W/O into O/W type (known as phase inversion of Pickering emulsions [23]) at very high phytosterol concentrations (e.g., 80 g/L, No. 5). In the O/W Pickering emulsions, no bacterial cells were observed in the continuous water phase (Figure 4B). Corresponding to the various structures of the Pickering emulsions, the accumulation of AD and ADD also changed dramatically (Figure 4C). The product concentration increased with the increase in the phytosterol concentration to 25 g/L and then decreased to the minimum value at the phytosterol concentration of 40 g/L. Very interestingly, the product concentration increased with the further increase in the phytosterol concentration above 40 g/L. The highest product concentration (4.2 g/L) was achieved at the phytosterol concentration of 80 g/L, which provided a method for microbial transformation with high phytosterol loadings.

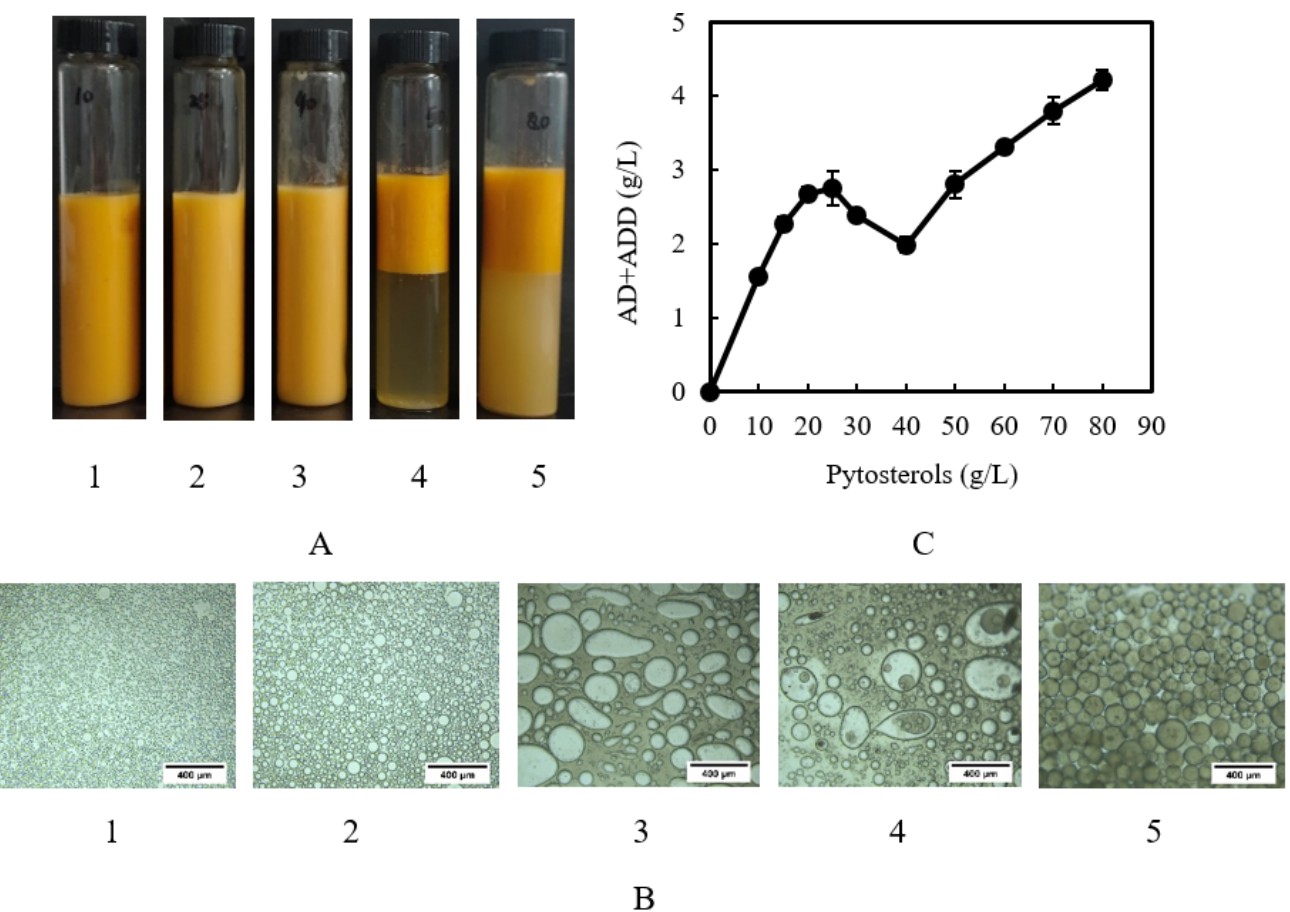

**Figure 4.** Microbial transformation with various phytosterol loadings: (**A**) Emulsion fraction. (**B**) Microscopic emulsions (bar = 400 μm). (**C**) AD and ADD concentrations. In the basic experimental condition, the volume ratio of soybean oil to the aqueous culture medium was 15:35 mL. The glucose concentration in the aqueous culture medium was 10 g/L. The microbial transformation time was 7 days. No. 1, 2, 3, 4, and 5 corresponded to phytosterol concentrations in the soybean oil of 10, 25, 40, 50, and 80 g/L, respectively.

### 2.5. Time Course of Microbial Transformation

The time course of microbial transformation with high phytosterol loadings was further examined (Figure 5). At the beginning of the microbial transformation, a W/O Pickering emulsion was observed (No. 1), and it had already been confirmed that the Pickering emulsions were stabilized by phytosterol crystals as solid emulsifiers [16]. Very recently, phytosterol crystals have also been utilized as emulsifiers for the fabrication of eatable W/O Pickering emulsions in the food field [24]. With the progress of microbial

transformation, the volume of the excess water phase increased and the flowability of the Pickering emulsions improved. At the same time, some bacteria–phytosterol crystal microaggregates were also observed in the excess water phase (No. 2–5) (Figure 5A). The microscopic morphology demonstrated that the shape of the water droplets in the continuous oil phase became irregular, and some bacteria–phytosterol crystal microaggregates were partitioned in the continuous oil phase within 1–3 days (No. 1 and 2). On the 4th day, multiple O/W/O emulsions were observed, where bacterial cells and phytosterol crystals were significantly partitioned in the oil phases (No. 3). Phase inversion occurred on the 5th day, where the multiple emulsions turned into O/W Pickering emulsions (No. 4). The O/W Pickering emulsions were maintained until the end of the microbial transformation (the 9th day, No. 5) (Figure 5B). Corresponding to the formation of O/W Pickering emulsions, rapid accumulation of AD and ADD was found, and 4.8 g/L of AD and ADD was achieved on the 9th day (Figure 5C).

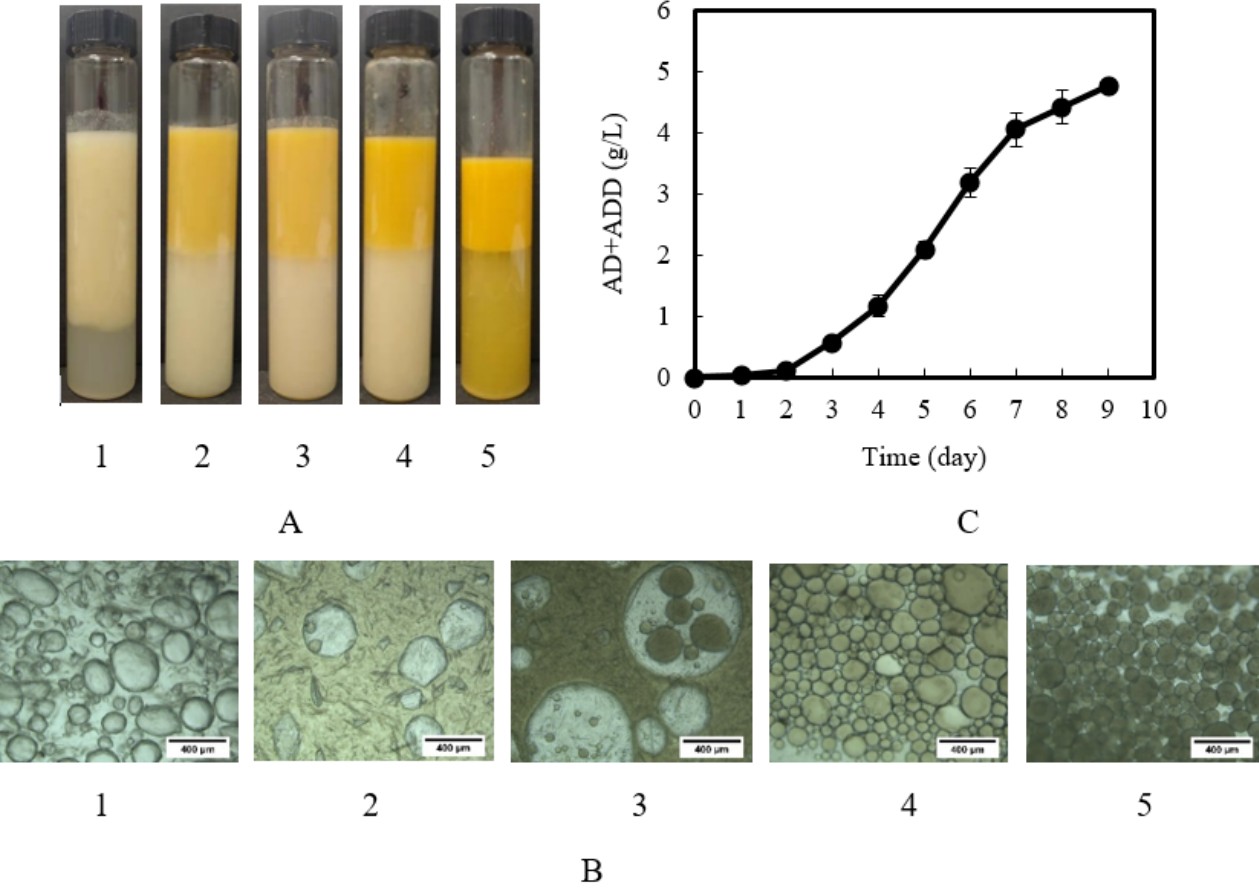

**Figure 5.** Time course of microbial transformation with high loading of crystal phytosterols: (**A**) Emulsion fraction. (**B**) Microscopic emulsions (bar = 400 μm). (**C**) AD and ADD concentrations. In the basic experimental condition, the volume ratio of soybean oil to the aqueous culture medium was 15:35 mL. The glucose concentration in the aqueous culture medium was 10 g/L. The concentration of phytosterols in the soybean oil was 80 g/L. No. 1, 2, 3, 4, and 5 corresponded to microbial transformation times of 1, 3, 4, 5, and 9 days, respectively.

## 3. Discussion

Pickering emulsions turning from W/O to O/W type, or vice versa, is called phase inversion [25,26]. Around the phase inversion point, Pickering emulsions usually present as multiple unstable emulsions with good flowability. Methods of adjusting phase inversion range from the co-stabilization of Pickering emulsions by adding other soft or hard particles [25,27], to the modification of solid particles' wettability by physical ad-

sorption [16,28,29], chemical graft modification [30,31], or even genetic manipulation of bacterial cells [32]. In addition to wettability, the concentration of solid emulsifiers is also a major factor influencing the phase inversion of Pickering emulsions [23]. Under the condition of high particle concentrations, self-aggregation causes the wettability of solid particles to change and phase inversion to occur [23,33,34]. It has also been reported that phase inversion of Pickering emulsions occurs with the increase in the inner phase fraction [35], and the corresponding demulsification effect is also applied for the downstream process of microbial fermentation [26]. In our previous work, we confirmed that hydrophobic *M. neoaurum* cells act as solid particles to stabilize W/O Pickering emulsions [15]. It has also been reported that phytosterol crystals act as solid particles to stabilize W/O Pickering emulsions [24]. Furthermore, adsorption of *M. neoaurum* cells onto phytosterol crystals to form crystal phytosterol–bacteria microaggregates, which has been confirmed by CLSM analysis, exhibits wettability unlike that of crystal phytosterol particles or bacteria [16]. The altered wettability of crystal phytosterol–bacteria microaggregates leads to the instability of Pickering emulsions. In the present work, the irregular shape of water droplets, multiple emulsions, and phase inversion of Pickering emulsions were observed with the increase in the phytosterol concentration, due to the altered wettability of the crystal phytosterol–bacteria microaggregates (Figure 4B). Vice versa, a similar phenomenon was also observed with the increase in the bacterial cell concentration during microbial growth (Figure 5B). Very high cell concentrations, as was achieved by the addition of high concentrations of glucose as a co-substrate, may lead to the formation of irregularly shaped water droplets or even multiple emulsions (Figure 3B). This may share the same mechanism of self-aggregation resulting from the high concentration of solid particles [23,33,34]. However, only an irregular shape of water droplets was observed at a very high volume ratio of oil to water (Figure 2B). Although high cell concentrations (caused by the high fraction of soybean oil; Figure 1A) favored the phase inversion from W/O to O/W, this trend may be counteracted by the high volume fraction of the inner phase (the water phase, not the soybean oil phase) in the W/O Pickering emulsions (favorable to the phase inversion from W/O to O/W [26]).

Hydrophobic solid particles act as emulsifiers of Pickering emulsions by attaching to the oil–water interfaces. When the solid particles are functionalized with catalytic activity, chemical reactions occur at the oil–water interfaces, and the solid particles act as both catalysts for the chemical reactions and solid emulsifiers for the Pickering emulsions, known as interfacial catalysis [36–38]. *M. neoaurum* cells catalyze the microbial transformation of phytosterols into AD and ADD (Figure 1B). At the same time, hydrophobic *M. neoaurum* cells can stabilize Pickering emulsions in a water–BEHP two-phase system [15,16] as well as in a water–soybean oil two-phase system (Figure 2A). In this case, interfacial biocatalysis provides a novel strategy for microbial transformation of hydrophobic chemicals, such as cholesterol [15] and phytosterols [16]. In the present work, we further confirmed the correlation relationship between the morphological structures of Pickering emulsions and the efficiency of microbial transformation. Around the phase inversion point of Pickering emulsions (No. 4 of Figure 4 with phytosterol concentration 40 g/L), multiple unstable emulsions with good flowability were observed, and the microbial transformation exhibited the lowest AD and ADD concentrations. With phytosterol concentrations below 40 g/L, the AD and ADD concentrations increased and then decreased with the increase in the phytosterol concentration, reaching a maximum volume of 2.8 g/L for AD and ADD at a phytosterol concentration of 25 g/L. This phenomenon is known as crystal substrate inhibition. The mechanism behind this phenomenon can be attributed to the unstable Pickering emulsions due to the formation of crystal phytosterol–bacteria microaggregates [16]. At phytosterol concentrations above 40 g/L, relatively stable O/W Pickering emulsions were formed, and the microbial transformation maintained a relative high activity. In addition, the crystal phytosterol–bacteria microaggregates in the excess water phase (Figure 4A) may also provide an additional route for microbial transformation of phytosterols via direct interaction between phytosterol crystals and bacteria [39]. In this way, efficient microbial

transformation with a high phytosterol loading becomes possible. At the same time, bacterial cell concentration is also an important factor affecting the morphology of Pickering emulsions (Figures 2 and 3). Thus, process parameters—such as glucose concentration [40] or the volume ratio of oil to water [5,19–21,41]—also strongly affect the microbial transformation of phytosterols in Pickering emulsions. Under the condition of a relatively high phytosterol loading, the time course of microbial transformation indicated that AD (ADD) accumulation was achieved at a relatively high rate (Figure 5). Microbial transformation of phytosterols involves the cascade processes of substrate phytosterol transport in the extracellular medium, transport of phytosterol across the cell membrane, and enzymatic catalysis in the intracellular environment. It should be pointed out that engineering of *Mycobacterium* sp. for enhancing transport of phytosterol across the cell membrane as well as the enzymatic degradation of phytosterols is only efficient [42,43] when phytosterol transport in the extracellular medium is not a rate-limited step.

## 4. Materials and Methods

### 4.1. Materials

*Mycolicibacterium neoaurum* (China Center of Industrial Culture Collection, CICC 21097)—a strain for accumulation of the steroid synthon AD by microbial transformation of sterols—was used in this study. Soybean oil (the major fatty acid constituents of which are C16:0, C18:1, and C18:2) was purchased from a local supermarket (Shanghai, China). AD and ADD from Sigma-Aldrich (St. Louis, MO, USA) were utilized as standard substances for HPLC analysis. Phytosterols (major components including β-sitosterol 40%, campesterol 20%, stigmasterol 17%, and brassicasterol 5%) from pilot plants were utilized as substrates for microbial transformation to produce the steroid synthons AD and ADD. Other chemicals were of analytical grade.

### 4.2. Microbial Transformation

*M. neoaurum* was maintained on a nutrient agar plate (10 g of yeast extract, 10 g of peptone, 10 g of glucose, and 15 g of agar per liter of water) at 4 °C. An isolated colony from the agar plate was inoculated into 50 mL of inoculum medium, i.e., 15 g of glucose, 5 g of $NaNO_3$, 5.4 g of $K_2HPO_4$, 2.6 g of $KH_2PO_4$, and 0.5 g of $MgSO_4·7H_2O$ per liter of tap water, and an initial pH of 7, cultured in a 250 mL Erlenmeyer flask at 200 rpm and 30 °C for 3 days.

The aqueous culture medium consisted of 10 g of $NaNO_3$, 5.4 g of $K_2HPO_4$, 2.6 g of $KH_2PO_4$, 0.5 g of $MgSO_4·7H_2O$, 0.1 g of $CaCl_2$, and 0.05 g of $FeCl_3·6H_2O$ per liter of tap water, in the presence or absence of 10 g/L glucose as a co-substrate. Next, 15 mL of soybean oil with 25 g/L phytosterols (no phytosterol crystals were observed), 35 mL of aqueous culture medium, and 2 mL of inoculum were combined in a 250 mL Erlenmeyer flask, which was incubated at 30 °C and 200 rpm for microbial transformation for 7 days (unless otherwise specified). During the microbial transformation, the water–soybean oil two-phase system turned into Pickering emulsions due to the growing bacterial cells acting as solid-particle Pickering emulsifiers.

### 4.3. Analysis of AD and ADD Concentrations

After microbial transformation, the whole transformation medium was also used for the analysis of AD and ADD concentrations. First, 30 mL of ethyl acetate was added to each tube and shaken at 200 rpm for 1 h. The AD and ADD, residual phytosterols, and soybean oil were extracted into the ethyl acetate layer. The total mixture was centrifuged at 10,000 rpm for 10 min. Aliquots of the ethyl acetate layer were withdrawn for AD and ADD analysis.

Aliquots (400 μL) of the ethyl acetate layer (containing AD and ADD, phytosterols, and soybean oil) were placed in a 70 °C oven to completely evaporate the ethyl acetate. The residues (oil and solids) were extracted 4 times with 1 mL of aqueous methanol solution (80%, *V*/*V*). All of the extracts of each sample were combined for HPLC analysis, which was

performed using a Shimadzu LC-20AT system (Shimadzu, Kyoto, Japan) equipped with an InertSustain C18 column (250 × 4.6 mm, 5 μm). The column was eluted with methanol and water (80:20, $V/V$) at a flow rate of 1 mL/min. The major products of microbial transformation were AD and ADD. The mass ratio of AD to ADD was approximately 5:1. AD and ADD were detected at 254 nm with retention times of 4.4 and 5.2 min, respectively. The corresponding concentration of the products AD and ADD was achieved based on the whole volume of microbial transformation.

### 4.4. Estimation of Microbial Growth

A certain volume of soybean oil (2, 4, 6, or 8 mL) was added to 40 mL of the aqueous culture medium (in the absence of glucose) as the sole carbon source. Then, 2 mL of inoculum was added, which was incubated at 30 °C and shaken at 200 rpm for 6 days for microbial culture. After the microbial culture, 30 mL of ethyl acetate was added to the broth, which was transferred into a 100 mL tube and centrifuged at 10,000 rpm for 10 min. The sediments in the aqueous solution phase were collected and dried at 70 °C for over 24 h until to a constant weight, before being utilized for the estimation of biomass.

### 4.5. Investigation of Pickering Emulsions

After microbial transformation, the Pickering emulsions and some excess water were transferred into a glass tube (80 mL) and stood for 2 h to allow for complete phase separation, which could be separated into an emulsion phase and an excess water phase. The phase separation system was captured using a digital camera. The emulsion fraction was deduced from the volume of the emulsion phase.

The emulsion type was inferred from the drop test, i.e., dilution of a drop of emulsion in either pure oil or water. The water-in-oil (W/O) emulsion was characterized by droplets that dispersed rapidly in oil while remaining in an agglomeration in the water, or vice versa for the oil-in-water (O/W) emulsion [44]. A small amount of the emulsion phase was picked up by a pipette and evenly spread on a glass slide, and microscopic images of the emulsion droplets were captured using an optical microscope (Olympus BX53M, Shinjuku, Tokyo, Japan) equipped with a high-speed CMOS camera (Basler acA2440-35uc, Ahrensburg, Germany).

## 5. Conclusions

Phase inversion of Pickering emulsions occurs during microbial transformation in the presence of a high crystal phytosterol loading. Efficient microbial transformation at high phytosterol loadings was achieved, where stable O/W Pickering emulsions were formed. The results confirmed the correlation relationship between the emulsion structures and the biocatalytic activity of the bacterial cells. In addition, process parameters such as the glucose concentration and the volume ratio of oil to water should also be optimized during microbial transformation, due to the influence of bacterial cell concentrations on the structures of Pickering emulsions.

**Author Contributions:** W.Z. performed the experiments and data analysis as well as writing the manuscript. H.X. assisted in the experiments and data analysis as well as the writing of the manuscript. X.Z. and Z.W. conceived and reviewed the manuscript. All authors have read and agreed to the published version of the manuscript.

**Funding:** The financial support from the National Natural Science Foundation of China (No: 22178218) is acknowledged.

**Data Availability Statement:** Data sharing is not applicable to this article as no datasets were generated or analyzed during this study.

**Conflicts of Interest:** The authors declare no competing financial interest.

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
