# Peer review of "Correlation Relationship between Phase Inversion of Pickering Emulsions and Biocatalytic Activity of Microbial Transformation of Phytosterols"

_catalysts, doi:10.3390/catal13010072_

Round 1
Reviewer 1 Report
In this research, by setting microbial transformation of phytosterols in a water-soybean oil two-phase system as an example, with the new viewpoint of interfacial biocatalysis in Pickering emulsions, Pickering emulsions turning from water-in-oil (W/O) emulsions into oil-in-water (O/W) ones, was observed during microbial transformation in the presence of high concentration of crystal phytosterols. However, it’s worth noting that some points need to be discussed and modified in the text. Hence, some specific comments are given in the following:
1) From Fig 2B, the author thinks that the microscopic morphology of Pickering emulsions shows the homogeneous droplets at low volume ratio of oil to water (No. 1-3), while the heterogeneous droplet sizes with some irregular big droplets were observed at the high volume ratio of oil to water (No. 4-5). Actually, all of the microscopic morphology of Pickering emulsions (No. 1-5) showed the heterogeneous droplet sizes with some irregular droplets. The author needs deal with the data again and give the explanation about why the further increase of oil volume (20:30, 25:25) led to the decrease of AD and ADD concentration.
2) From 2.1 and 2.2, the author described objectively that with phytosterols added, the water-soybean oil two-phase system turned into W/O Pickering emulsions, and the volume of Pickering emulsions increased with the increasing soybean oil volume. However, the author does not explain the phenomenon and analyze who act as the emulsifiers.
3) The author said that the glucose was influenced on Pickering emulsion structures (W/O turning into O/W/O), which need analysis about the function of the glucose in detail.
4) From 2.4, when increasing phytosterol concentration(>40 g/L), some micro-aggregates of phytosterol crystals were observed. Does the amount of phytosterol crystals increas with the increasing phytosterol concentration? In the previous work from the the author´s group (Applied Microbiology and Biotechnology (2022) 106:2403–2414): decreasing phytosterol degradation with the increase loading of crystal phytosterols. The Fig 4c showed different results, please comment on it.
5) The author confirmed that the hydrophobic M. neoaurum cells and phytosterol crystals both can act as solid particles to stabilize W/O Pickering emulsions. Then the author indicated that the crystal phytosterol-bacteria micro-aggregates leads to the instability of Pickering emulsions, which need detailed discussion. Furthermore, how can the phase of Pickering emulsions inversion from W/O to O/W with the crystal phytosterol-bacteria micro-aggregates?
Author Response
In this research, by setting microbial transformation of phytosterols in a water-soybean oil two-phase system as an example, with the new viewpoint of interfacial biocatalysis in Pickering emulsions, Pickering emulsions turning from water-in-oil (W/O) emulsions into oil-in-water (O/W) ones, was observed during microbial transformation in the presence of high concentration of crystal phytosterols. However, it’s worth noting that some points need to be discussed and modified in the text. Hence, some specific comments are given in the following:
1. From Fig 2B, the author thinks that the microscopic morphology of Pickering emulsions shows the homogeneous droplets at low volume ratio of oil to water (No. 1-3), while the heterogeneous droplet sizes with some irregular big droplets were observed at the high volume ratio of oil to water (No. 4-5). Actually, all of the microscopic morphology of Pickering emulsions (No. 1-5) showed the heterogeneous droplet sizes with some irregular droplets. The author needs deal with the data again and give the explanation about why the further increase of oil volume (20:30, 25:25) led to the decrease of AD and ADD concentration.
In generally, the sum of droplets with an irregular morphology are relatively large at high oil fraction. The picture is difficult to see it. However, the droplets are heterogeneous while sizes of droplet are changed. Those changes are easily observed, which had been corrected.
2. From 2.1 and 2.2, the author described objectively that with phytosterols added, the water-soybean oil two-phase system turned into W/O Pickering emulsions, and the volume of Pickering emulsions increased with the increasing soybean oil volume. However, the author does not explain the phenomenon and analyze who act as the emulsifiers.
The bacteria act as emulsifiers. Volume ratio is the factor affecting the emulsion type, W/O or O/W.
3. The author said that the glucose was influenced on Pickering emulsion structures (W/O turning into O/W/O), which need analysis about the function of the glucose in detail.
Glucose acting as co-substrate improves the bacterial growth, and then the bacterial cells act as emulsifier, which had been explained in discussion section
4. From 2.4, when increasing phytosterol concentration(>40 g/L), some micro-aggregates of phytosterol crystals were observed. Does the amount of phytosterol crystals increas with the increasing phytosterol concentration? In the previous work from the the author´s group (Applied Microbiology and Biotechnology (2022) 106:2403–2414): decreasing phytosterol degradation with the increase loading of crystal phytosterols. The Fig 4c showed different results, please comment on it.
In our previous works, microbial transformations were carried out at a relatively low phytosterol loading, i.e., within 40 g/L. The present work was carried out at a very high phytosterol loading and then new fact of high bioactivity was observed. Those facts are discussed in the discussion section.
5. The author confirmed that the hydrophobic M. neoaurumcells and phytosterol crystals both can act as solid particles to stabilize W/O Pickering emulsions. Then the author indicated that the crystal phytosterol-bacteria micro-aggregates leads to the instability of Pickering emulsions, which need detailed discussion. Furthermore, how can the phase of Pickering emulsions inversion from W/O to O/W with the crystal phytosterol-bacteria micro-aggregates?
The crystal phytosterol-bacteria micro-aggregates leads to the instability of Pickering emulsions, which had been detailed studied and discussed in our previous work (reference 16). The phase of Pickering emulsions inversion from W/O to O/W at high phytosterol loading is the content of present work, which should be attributed to the high solid emulsifier concentration, which had been discussed in discussion section.
Reviewer 2 Report
Ref. No.: catalysts-2071599
Manuscript: Correlation between phase inversion of Pickering emulsions and catalytic activity of microbial transformation of phytosterols
Journal: Catalysts
Some comments are as follows:
1. Lines 50-51 “we find a novel mechanism for M. neoaurum to utilize phytosterols solubilized in the BEHP (bis-(2-ethylhexyl) phthalate) phase of a water-BEHP two-phase system” A novel mechanism? Avoid the use of we.
2. Line 59 “bacterial cells stabilized Pickering emulsions”? Can bacteria stabilize the emulsions?
3. Line 84-85 “we tried to explore a new strategy for microbial transformation with high phytosterols loadings.” New? The authors should explain more in detail about this new starategy?
4. Line 302 The composition of “Soybean oil” should be provided.
5. Line 316 “15 ml soybean oil with 25 g/L” ml or mL? Please unify.
6. Lines 319-321 How is it converted to emulsion? What are the emulsification conditions?
7. Lines 116-117 “The Pickering emulsions became very viscous with the increase of soybean oil volume” At present, there is no result of viscosity increase, it is recommended to add rheological description.
8. Lines 119-120 “However, the flow-ability of Pickering emulsions improved markedly at the high volume ratio of oil to water.” How to prove?
9. The difference in the color of serum layer in the bottom should be explained.
Author Response
Some comments are as follows:
- Lines 50-51 “we find a novel mechanism for M. neoaurum to utilize phytosterols solubilized in the BEHP (bis-(2-ethylhexyl) phthalate) phase of a water-BEHP two-phase system” A novel mechanism? Avoid the use of we.
It was corrected.
- Line 59 “bacterial cells stabilized Pickering emulsions”? Can bacteria stabilize the emulsions?
Yes. This content had been confirmed in our previous work (reference 15)
- Line 84-85 “we tried to explore a new strategy for microbial transformation with high phytosterols loadings.” New? The authors should explain more in detail about this new starategy?
It was corrected.
- Line 302 The composition of “Soybean oil” should be provided.
The major fatty acid constituents of soybean oil was C16:0, C18:1 and C18:2, which was added
- Line 316 “15 ml soybean oil with 25 g/L” ml or mL? Please unify.
Thanks
- Lines 319-321 How is it converted to emulsion? What are the emulsification conditions?
During microbial transformation, Pickering emulsion formed. The content was modified
- Lines 116-117 “The Pickering emulsions became very viscous with the increase of soybean oil volume” At present, there is no result of viscosity increase, it is recommended to add rheological description.
It was observed directly while no datum shown, which the content was added
- Lines 119-120 “However, the flow-ability of Pickering emulsions improved markedly at the high volume ratio of oil to water.” How to prove?
Direct observed by shaken flasks
- The difference in the color of serum layer in the bottom should be explained.
The color was the un-identified co-metabolite of microbial transformation